# Needs Assessment in Parents of Children Affected by Cancer: A Qualitative Perspective

**DOI:** 10.3390/children9121957

**Published:** 2022-12-13

**Authors:** Blanca Bretones Nieto, Carmen Pozo Muñoz, María Ángeles Vázquez López

**Affiliations:** 1Department of Psychology, University of Almeria, 04120 Almeria, Spain; 2Maternity and Children’s Hospital, Torrecárdenas University Hospital, 04009 Almeria, Spain

**Keywords:** childhood cancer, parents, adaptation, needs, psychosocial aspects (cancer), qualitative methodology

## Abstract

Background: Childhood cancer is a “stressful experience” for parents in their role as caregivers. The aim of this study is to analyze the needs of a group of parents who have children diagnosed with cancer. The assessment looks at all areas of their daily life using a qualitative content analysis approach. Methods: This study uses open questions designed to investigate the main problems faced by the parents of children with cancer. All the answers were analyzed using MAXQDA 20 software. The study was conducted according to the Consolidated Criteria for Reporting Qualitative Research (COREQ). Results: The content analysis of the answers from 13 mothers and 7 fathers is presented. The main themes as priority areas of need were: “informal social support”, “coping”, “stressors/imbalances” and “health problems”. Dissatisfaction with the support provided from the informal network was detected. Although the participants expressed active coping, they also exhibited helplessness, as well as concerns regarding possible sequelae and the impact of this situation on the family’s economic/work context. Conciliation problems were a source of stress. All this is consistent with the participants suffering from a variety of health issues. Conclusions: the results show the effect of childhood cancer on different aspects of family life. They provide essential information for designing psychosocial interventions.

## 1. Introduction

According to the World Health Organization, an estimated 400,000 children and adolescents aged 0–19 years old develop cancer around the world each year [1]. In Spain, 12,183 cases of childhood cancer were registered among the 0- to-14-year-old population between 2010 and 2021 [2], which gives us a quantitative picture of the importance of this problem.

Childhood cancer is a highly stressful event for the family, affecting their physical and mental health, and causing extreme distress and trauma for the parents [3,4]. From the moment the child’s illness is diagnosed, all family members need to adapt to the new circumstances [5]. In general, it is the parents who perceive the greatest impact on their family life. This is motivated, among other reasons, by their role as primary caregivers [6,7,8,9]. This impact seems to be more apparent in mothers. Some studies show that mothers, as primary caregivers, exhibit more unsatisfied needs related to health issues, family social support and, especially, practical or instrumental social support [10,11].

Previous research has also shown an association between parental distress and the child’s functioning, even after the child has overcome his/her illness. Specifically, it has been shown that parents of children with cancer have a greater prevalence of anxiety, depression and post-traumatic stress disorder (PTSD) [4,12].

Nevertheless, existing research has mainly focused on minors, with minimal attention being given to the psychosocial impact the disease has on their parents, who provide the main source of support for their sick children. Consequently, investigating these problematic situations, and taking action to ameliorate them, is an indirect way of caring for the minors themselves [13].

Although there are many studies aimed at identifying the psychosocial repercussions on the parents of affected children, only very recently have they begun to look at the psychosocial needs of parents from the moment of diagnosis onwards, going through the different stages of the child’s disease [14].

### Areas of Need and the Most Significant Issues for Parents of Children with Cancer

Systematic needs assessment is a relevant first phase for gaining an in-depth understanding of the areas in the parents’ lives that have been affected by their child’s disease [15]. It allows us to identify the problems and shortcomings in a group, analyzing the different areas affected, prioritizing them and assessing the social and personnel resources, as well as their possible solutions. For this reason, the needs assessment model of Hernandez, Pozo and Alonso has been followed [16,17,18,19] in this work. The model consists of two phases: (1) identifying areas of need and (2) describing and prioritizing existing needs in previously identified areas (paying special attention to the affected population’s own perspective). One virtue of this proposal is its ability to provide detailed, extensive, contextualized and in-depth descriptions of existing problems and needs, thus facilitating their understanding.

An in-depth understanding of the needs of parents can help in designing interventions that address the problems and deficiencies detected and mitigate the impact of these factors on their health and well-being. It also helps them to cope better with psychosocial stress, which, ideally, will eventually benefit the child by providing more positive support for the family [11].

Analyzing the parents’ speech content facilitates the gathering of information, allowing us to understand the situations experienced from their personal perspective [20]. Not only are the parents’ health and well-being affected by childhood cancer, but various areas of their daily functioning undergo important changes that affect the quality of the entire family’s life [21,22]. Having to reduce working hours to accompany the child during their treatment, economic difficulties, poor and sometimes almost non-existent relationships with friends, relationship problems, and scant attention paid to the rest of the family are among some of the most evident consequences [3].

During the disease process, parents require information and guidance, especially from health professionals. They demand to have their questions answered, to be informed of the procedures administered and to be made aware of when the expected side effects will occur [23]. Medical and physical health information has been identified as the most important concern for parents. Psychoeducational care is also essential, with a multidisciplinary team that includes psychological and social care [24].

In addition to informational support, parents commonly require social support, both informal—from family members, including grandparents and siblings [25]—and formal (health personnel, bosses, etc.). It has been demonstrated that perceived social support provides health and well-being benefits [19,26].

Another area that is particularly affected is the health of the parents themselves, especially once the child’s treatment has ended, and even up to several years afterwards. The experience causes psychological distress in the parents and a higher prevalence of anxiety, depression and post-traumatic stress disorder (PTSD) compared to the control populations [27,28]. This is because the parents must continually deal with multiple sources of stress, such as waiting for medical results, uncertainty or helplessness when receiving the prognosis and witnessing the child’s suffering, etc. [3]. Hence, knowing the parents’ main “fears and concerns” has become a primary objective for future interventions [29].

At the same time, parents are aware of their deteriorating economic situation, caused by unforeseen expenses and a reduction in income [30]. The economic burden placed on families during the child’s treatment has been shown to have considerable long-term consequences on financial security [31]. In extreme situations, families can experience significant economic hardship, even poverty [1]. A difficult economic situation such as this can further deteriorate when the parents, who act as caregivers for their children, need to reconcile their intra-hospital situation with their work, often having to interrupt their working day or resign from their job; this is particularly the case for mothers [32].

As soon as treatment begins, there are further obstacles to the day-to-day functioning of family life. There is a decrease in social and interpersonal relationships since attention is focused on caring for the sick child and the immediate family; the parents deprive themselves of much of their social life and ignore their own basic needs [33,34]. Marital problems also manifest, parental functioning is affected and communication difficulties arise [35].

Educational problems and falling behind at school are frequently observed (both during and after the disease). In addition, parents feel that their children may have difficulties in forming and maintaining relationships due to changes in their image, possible sequelae or prolonged absences from the school environment [36,37].

To resolve the day-to-day issues arising from their child’s illness, parents use coping strategies (cognitive, affective and behavioral processes). In this regard, existing studies agree that the strategies implemented by parents of children with cancer are not truly adaptive. They have shown a correlation between coping and somatic symptoms and that many parents use emotion-focused coping strategies rather than problem-focused coping strategies [38,39].

In short, in our work, we focus on “what is the experience like of having a child with cancer?” and we intend to answer the following three questions:-How does childhood cancer affect the parents of sick children?-In what areas do problems manifest most?-How do parents deal with such problems?

Ultimately, the purpose of this study is to assess the perceptions of parents who are affected by childhood cancer and to identify their needs from a qualitative perspective, taking into consideration all areas (a comprehensive needs assessment) of their day-to-day life.

## 2. Materials and Methods

### 2.1. Study Population

This study is part of a broader investigation, Psychosocial Repercussions of Childhood Cancer in Parents, which combines quantitative and qualitative methodologies. The current paper presents the qualitative parental study (a subsample drawn from the larger study) using semi-structured interviews (see Table 1) to gather in-depth information on relevant areas of need, information that is impossible to extract using the usual quantitative methods.

The interviewees were mothers (*n* = 13) and fathers (*n* = 7) of children and adolescents diagnosed with cancer and receiving active treatment in a hospital in southern Spain (the Pediatric Oncology Unit). With regard to the disease, as an inclusion criterion, it was agreed that it was important for the sick child to be in the “active treatment” phase (i.e., receiving cycles of chemotherapy).

### 2.2. Study Design and Data Collection

The reporting of this study was conducted according to the Consolidated Criteria for Reporting Qualitative Research (COREQ) [40]. This comprises a 32-item checklist that ensures the quality of the interviews. Specifically, the tool guides researchers on: the research team and reflexivity (personal characteristics and the relationship with participants), the study design (its theoretical framework, participant selection, setting and data collection), the analysis and the findings (data analysis and reporting). Unlike the way it is usually employed, we used this checklist at the beginning of the process as a guide for ensuring the quality of the qualitative research being carried out.

The study was approved by the hospital’s Research Ethics Committee and was conducted in accordance with the 1975 Helsinki Declaration (as revised in 2013) [41].

Based on an exhaustive literature review, a list of 10 topics constituting potential areas of need in parents of children with cancer was devised [3,10,20,23,24,25]. From there, ad hoc interview questions were considered (see Table 1).

Before conducting the interview, the parents were informed of its characteristics, purpose, how the results would be used and data confidentiality. All of the participants were required to sign a statement of informed consent.

All the interviews were conducted by members of the research team, who are expert researchers in social psychology and qualitative research. Each interview lasted 40–60 min. All interviews were recorded and transcribed verbatim. Strategies were established to avoid bias during the data collection process (decreased attention caused by tiredness, a transcription bias or a probing bias). For example, a maximum limit of four interviews per day was established and neutral phrases were used.

### 2.3. Data Analyses

The transcripts were analyzed through qualitative content analysis based on the approach by Mayring using MAXQDA software [42,43]. Themes were generated deductively from the interview guide. Other categories were derived inductively from the transcript data.

Coding systems were consolidated and revised, and a final coding guide was developed that included category labels (theme/subtheme) and category definitions (“meanings”) [see Table 2]. All the interviews were coded based on this final coding system.

The results were discussed, agreed upon and validated by all the researchers. Sociodemographic variables and clinical characteristics were analyzed and described with IBM SPSS Statistics V.25 software.

## 3. Results

### 3.1. Sample Characteristics: Sociodemographic and Clinical Variables

Most of the participants were of Spanish nationality (84%), aged between 22 and 69 years (mean = 42.05; sd = 7.65); 88.4% were married or living together as a couple, 60.2% had completed secondary education and half (53.9%) worked in the service sector. Regarding the children’s diagnosis: 44.2% were diagnosed with leukemia and 55.8% with solid tumors (to differentiate between the latter and hematological diseases).

### 3.2. Themes

The results show that five of the ten needs raised during the interviews are major problems for the study participants. The five main themes identified (the priority areas of need) were: “informal social support”, “coping”, “fears and worries”, “stressors/imbalances” and “health problems in parents”. In addition, other topics of parental concern were found but obtained a lower presence in the discourse; these were: the children’s education, formal support and social resources and work/economic problems. Within each area, we have specified the most significant parental concerns based on their own perceptions. Table 2 shows the results obtained using MAXQDA software. Below are some of the parents’ testimonies taken directly from the interviews under the five emerging themes and subthemes, and their meanings.

#### 3.2.1. Theme 1: Informal Social Support

Respondents identified various sources of support in their statements. The main ones were: the spouse, friends, neighbors and co-workers.

Concerning the spouse, the experience of illness in the child causes discomfort between the couple. The participants believe that their relationship has deteriorated since the beginning of their child’s illness. Added to this situation is the psychological impact of the child’s illness on a personal level.

In this regard, one father said:


*“I don’t know what I want to do, I just want to spend as much time as possible with my children… If I didn’t have my children I would live alone, I wouldn’t need to be with my wife…”*

*(Father, Participant 6)*


Likewise, mothers show difficulties in interacting with their husbands over the course of the disease:


*“My partner does not listen to me much and rather discourages me. I make the decisions because he is very indecisive. He doesn’t do anything in the house when I’m with the girl. If I’m here for five days, he won’t sweep for five days”*

*(Mother, Participant 10)*


Similarly, with regard to the role of friends, neighbors and colleagues, some parents were disappointed by people they considered important to them:


*“I have friends who’ve been absent since my daughter got sick”*

*(Mother, Participant 12)*


Whatever the source of support, it is important to have of a wide social network to strengthen interpersonal relationships. However, the daily situation caused by the child’s illness, as well as feelings of sadness and distress, lead, in some cases, to the parents becoming isolated:


*“I lost many friends because I locked myself in and didn’t go out”*

*(Mother, Participant 13)*


Although the social support provided can be very useful for parents, it is not always helpful because it does not meet their needs at certain times. Therefore, they found the help they were getting dysfunctional, not giving them support:


*“The neighbors keep asking about my daughter and that doesn’t support me. They do not understand the problem or the seriousness of the issue”*

*(Mother, Participant 10)*


In short, perceived social support and, more specifically, satisfaction with it, has proven to be a fundamental element for the parents. The different arguments expressed by mothers and fathers have insisted on this issue as being important.

#### 3.2.2. Theme 2: Coping

Many studies have contrasted the influence of different forms of coping on psychosocial wellbeing [44,45,46].

Among the parents interviewed, there were many allusions made to helplessness (the first subtheme in this category). The participants had difficulty managing their emotions with regard to the lack of control they had over the disease:


*“Seeing how she suffers every day and not being able to do anything…… I don’t like to live with uncertainty. I am overwhelmed by the feeling of not knowing and suffocating. I feel helpless in the face of what is happening to my daughter”*

*(Mother, Participant 20)*


This uncertainty and the inability to cope with the situation are also manifested when they refer to the treatment, the suffering of their children or the post-treatment repercussions:


*“It makes me feel impotent and angry that my daughter can’t move. She still depends on her sister to help her…”*

*(Father, Participant 3)*


Helplessness also manifests in the face of the uncertainty of having made (or not made) good decisions regarding your child’s illness. Thus, parents said:


*“The situation drags you along and doesn’t let me decide how to deal with it. I’m afraid I was wrong…”*

*(Father, Participant 6)*


Nevertheless, they also demonstrate active coping (the second coping subtheme), which focuses on decision-making and emotional self-regulation as mechanisms to ensure the well-being of the sick child and the rest of the family. Mothers in particular indicated that they have played a leading role in the family crisis caused by the child’s illness:


*“I am the support in my house. My children and my husband turn to me. Since my daughter became sick, I make all the decisions”*

*(Mother, Participant 14)*


Finally, the last subtopic extracted from the MAXQDA software regarding the parents’ statements brings together the answers concerning the parents’ negative coping with their child’s illness:


*“When you think about the future. Puff…, that’s the worst time of the day… There are days when you think all the time that you can die… I never think about anything positive”*

*(Mother, Participant 20)*


#### 3.2.3. Theme 3: Fears and Concerns

Parental responses to this topic were divided into two main subthemes: (1) fears related to the disease’s sequelae and possible relapses and (2) concerns about possible economic and work problems.

The child’s health was the most upsetting aspect for the parents, in conjunction with worrying about sequelae and relapses:


*“The main thing is to heal and not have any sequelae”*

*(Father, Participant 9)*



*“… a possible relapse worries me very much … the repercussions that chemotherapy will have on his health in the future…”*

*(Mother, Participant 11)*


They are also worried about the child’s future. Fears are intertwined with wishing the best for their sick child in the not-too-distant future (this is not treated as a subcategory, since it is understood as fear and concern about the child’s happiness and future. All this refers to their prognosis and the consequences mentioned above):


*“I want my daughter to be cured and to be happy…”*

*(Mother, Participant 7)*


However, the sick daughter or son is not the only concern. Parents are afraid that the situation caused by the illness will alter family stability. The other children also suffer from the child’s illness and the family as a whole is affected:


*“I am worried about my sick daughter, my wife, and my other daughter because she is lonely and struggling to cope…”*

*(Father, Participant 17)*


Another concern that systematically emerges in the parents’ transcripts is that related to the economic and work spheres. Some parents have had to travel with their child to other cities to receive care, some have asked to work less and others have even had to quit their job. For example, one mother said she was very concerned:


*“…that either of us could be out of a job”*

*(Mother, 11)*


On the other hand, one father argued:


*“In the workplace I am afraid of the economic uncertainty and that the company may close”*

*(Father, Participant 3)*


Although it was not originally treated as a study item, participants stated that the family situation caused by the disease had led them to restructure and prioritize their values and needs (not considered as a subcategory in Table 2. This is because they express it in their lives as a result derived from “fears and concerns”).


*“I would trade the time I took from my son (and my wife and family) for my job. I let something very valuable go. My son’s illness helped me change the “chip” and give importance to some things rather than others”*

*(Father, Participant 2)*


#### 3.2.4. Theme 4: Stressors/Imbalances

Childhood cancer is a stressor that affects the health and well-being of parents. What causes them most upset is having to cope with the demands associated with the illness (classified as the “disease claims” subtheme). One father said:


*“I feel a lot of impotence, anger, discomfort and indignation. Whatever I do, I don’t see any light. Whatever you do, it is very difficult to disconnect at all. My son’s illness is always on my mind… I am afraid of being wrong…”*

*(Father, Participant 6)*


Another stressor is the difficulty in being present during the medical procedures that the child must undergo and that can be painful. One mother stated:


*“When she had the first lumbar puncture, I was in the room. Later I couldn’t bear it. When they prick her, I sometimes grab her because I suffer, I get tachycardia and can’t sleep; it’s all the result of seeing her screaming and crying”*

*(Mother, Participant 7)*


One of the most stressful aspects is trying to combine the demands associated with childhood disease and the family obligations. For example, one mother described these difficulties in reconciliation by saying:


*“Sometimes I had to prepare food at home for the rest of the family, then go to the hospital … I couldn’t do anything else”*

*(Mother, Participant 1)*


Dealing with the child’s illness continually overburdens parents, especially mothers; this can have negative consequences on their health and well-being:


*“There are days I get stressed, including double dates with doctors, car trips, after-school classes…”*

*(Mother, Participant 4)*


In short, combining the demands associated with childhood illness and family obligations, as well as managing the feelings associated with having to be present during medical procedures performed on the child, which can be painful, are among the most stressful aspects for parents [47].

#### 3.2.5. Theme 5: Health Problems in Parents

The stressors mentioned in the previous section may have a particular impact on parental health. Participants reported suffering from a wide variety of health problems during their child’s illness. Parents say they suffer from anxiety, nausea, stomach pains, feelings of sadness and depression, respiratory difficulties, exhaustion, concentration problems, headaches, migraines, dizziness, insomnia, lack of motivation or changes in self-esteem and personality. Parents talk about specific physical symptoms:


*“…the mornings are especially bad, my stomach hurts and I feel nauseous, until I get up and start the day”*

*(Mother, Participant 5)*


They are absorbed in their child’s care and harbor many recurring fears:


*“You live with fear, you’re not the same person as before… Everything is fear…”*

*(Mother, Participant 5)*


Furthermore, they resort to using medicines and practicing unhealthy habits. However, they do not seek help from professionals since they devote all their attention to their sick child.


*“…If I don’t take the pills, I’m so tired. I can’t stand up…”*

*(Mother, Participant 20)*


In short, the physical and psychological health of both parents are affected, but their child’s illness prevents them from taking care of themselves. A father summarizes very clearly the variety of symptoms and discomfort he suffers:


*“I have very low self-esteem and I don’t love myself very much. It’s like I have absorbed energy. I don’t feel like talking to anyone. It’s like I don’t have a life or motivation. I don’t laugh much. On the days I feel more stressed for my child, I am overwhelmed, feel exhausted, and have breathing difficulties. For the last year, I have had discomfort in both eyes. I dry them, especially at night. I think it is because I think too much … Sometimes I get nauseous without knowing why and I get migraines. I have to take anti-inflammatory drugs every day and they relieve the symptoms. This happens every two or three days… I also drink a lot of caffeine and smoke. I find myself physically and mentally weaker than 2 or 3 years ago…”*

*(Father, Participant 6)*


#### 3.2.6. Other Themes

In addition to the themes and subthemes recorded above, analyzing the parents’ statements has enabled us to detect other sets of problems and needs that they face throughout their child’s illness. We believe that these are also relevant.

Work and the Family Economy

Most parents have to take time off and/or leave their jobs in order to be temporarily available to care for their sick child. This group also experiences concentration and work performance problems:


*“I stopped working because of my son’s illness, because we moved to a hospital in another city. We were with my son for nine months after the first transplant. Since then, I haven’t been able to work”*

*(Mother, Participant 12)*


Economic problems resulting from the disease situation make the parents’ lives difficult: they consider their income as being insufficient; on numerous occasions, they have to quit their jobs, use their savings and even sell their property:


*“I have many financial and work problems; I don’t have the job I want, and I get paid very badly. My wife and I quit our jobs because we’ve been with the girl constantly for over two years, and we’re both unemployed”*

*(Father, Participant 3)*


2.Education of Children

The results, which relate to the evolution of learning processes and educational development, are quite heterogeneous. It is very common for children to suffer a decline in academic achievement. The following are some of the problems caused by school absenteeism, the detection of cognitive and/or attentional dysfunction derived from the effects of oncological treatments and the sequelae of cancer, as well as problems with social interaction.


*“She didn’t go to school for three years and attended little more than a month during the fourth year”*

*(Father, Participant 3)*



*“He doesn’t know how to relate well with other children, they are more active. In certain ways, he’s less mischievous”*

*(Father, Participant 6)*


3.Formal Support and Social Resources

Doctors, nurses and nursing assistants were fundamental support agents. The assessment of care received was very satisfactory, both in terms of the confidence and tranquility these support agents provided to the families and in terms of the information they gave to both the patient and the parents. Nevertheless, communication was more fluid with the nursing staff than with the doctors.


*“I valued the medical team a lot…. They were very reassuring”*

*(Father, Participant 2)*



*“Talking to the nursing staff made me less worried than talking to the doctors”*

*(Mother, Participant 16)*


This study also demonstrates the important role played by NGOs in the field of social assistance. In this case, a parent’s association working in the hospital and in an out-patient capacity:


*“They helped me psychologically. Now, they even call me and ask about my daughter. You feel that you’re not disoriented and adrift, and when I needed something, they helped me and gave me advice …”*

*(Mother, Participant 4)*


On the services offered by public institutions, opinions were quite unfavorable:


*“Now they’ve given us the disability card to park at the hospital. Before, we had to carry my daughter in our arms from the parking lot…. every day for weeks and months”*

*(Father, Participant 3)*


Appendix A expands the information with examples of the parents’ discourse, classified by themes and subthemes.

## 4. Discussion

This study examines the parents’ perception of their psychosocial needs following their child’s cancer diagnosis via a semi-structured interview. It is part of a larger study that also included previously validated scales [48], although this work presents exclusively the qualitative research part [3]. This article presents the qualitative results obtained from ad hoc questions about areas of daily life. For this, the COREQ method has been used, given that it is a tested procedure for efficiently conducting qualitative research [40]. From the content analysis carried out on the obtained categorizations, it is important to analyze the following aspects.

Regarding the informal system, Alonso, Menéndez and González [49] emphasize the role played by the family as a caregiving unit. It has been shown that receiving support from the people who make up the close family and/or social circle is beneficial for health and well-being, in this case, of parents who have children with cancer [49]. Nonetheless, the results show that the participants are not satisfied, for example, with regard to the perceived support from their partner.

Friends, neighbors and co-workers are also valued as providers of emotional and instrumental support. It would therefore be of interest to reinforce its usefulness as a strategy for better adapting parents to the different stages of their child’s illness.

Despite this, the support received by the informal network has not always been functional for the parents interviewed [8]. In some cases, a tendency toward social isolation was detected, an issue also exhibited in studies by other authors [50].

Similarly, the influence of different forms of coping on the psychosocial well-being of this population has already been illustrated [10]. In this regard, feelings of helplessness are common among the participants. This happens in parallel with the discomfort derived from coping with stress, including the tendency to practice unhealthy strategies or patterns.

At the same time, the participants were worried about elements related to the child’s health (and the possibility of relapse or sequelae as a result of the disease and treatments, and the well-being of siblings and the immediate family in general. It is worth noting how they link “health” with “happiness”. The participants even stated that this experience led them to change their personal values. Such findings replicate those of other authors [51,52].

With regard to stressors and family imbalances, what most disturbs the parents are the combined demands of the illness, daily obligations and a son or daughter struggling with painful medical procedures [10].

In the areas of work and household economy, parents face serious difficulties. These are areas that greatly concern them and which, in turn, are categorized as “Other themes”. Similar results have been observed in other studies. Bona et al. [53] reported that parents had to deal with continual interruptions in their work; in almost half of the cases, one of the parents had to leave their job and, in all, the child’s illness was a serious problem that affected them economically and resulted in a substantial loss of income.

In Spain, over recent years, public administrations have been trying to design strategies that provide support and cover for parents who have children with cancer. For example, a law has been implemented for the provision of care to minors who are affected by cancer or other serious illnesses [54]. This includes the allocation of resources for the caregiving parent and assistance in reconciling family and professional needs. Decree 154/2017 [55] has also been approved recently; this regulates permissions given to public servants so that they can care for children with cancer or other serious illnesses. To understand the results obtained, it is vitally important to continually evaluate the effectiveness of these laws for the purpose of redefining and adapting them to the problems and difficulties experienced by the parents themselves. In this context, the role of associations for affected patients must be highlighted since they provide the necessary advice to families and manage these types of social benefits. Regarding the services offered by such bodies, the parents in this study expressed favorable opinions.

Regarding “stressors and imbalances” and “health problems” (following the order set out in the Results section), the statements relate to the parents’ primary caregiver role. Here, they express how difficult it is to deal with the disease and the painful treatments. Nonetheless, because of their child’s vulnerability, they manifest “active cognitive coping”, in which they strive to resolve any problematic situations that arise and confront the difficulties resulting from the pathology. Thus, this pattern implicitly involves responses aimed at facing and resolving any possible demands and, in turn, reorganizing the rest of their daily responsibilities, relegating their own needs to the background [10,49,56].

Finally, we consider the “other themes” to be relevant. Although in this work we have concentrated on analyzing the effects of childhood illness on the economic and work spheres, the difficulty perceived by participants regarding “the education of their children” is also notable, as well as the role played by “formal support and social resources”.

With regard to how well the available social benefits function in Spain, a law has recently been approved that recognizes a disability level of 33% for children with cancer [57]. Its implementation should help lessen the demands made on parents. Nevertheless, its effectiveness should be evaluated in a few years to determine how well it responds to the difficulties faced by these children and their families.

The work carried out by healthcare professionals is also decisive in terms of support. The parents in this study were satisfied with the care provided by doctors and nurses. However, it was noted that communication with the latter is much more fluid than with the former. Many researchers highlight the need for communication between doctors and parents to go beyond the simple transmission of technical and prognostic information [58]. The doctor is responsible for creating a safe and trusting environment and the parents need their questions to be answered, to understand the procedures being administered and to have an idea of when the expected side effects will occur [23].

Therefore, comprehensive care systems should not only focus on medical aspects, but also prioritize the social sphere and cultural characteristics, etc. [59]. This is the working model that the public administration in Spain has been trying to optimize over recent years by integrating professionals from different disciplines, people who understand the reality and difficulties faced by children and adolescents diagnosed with cancer a model that also includes the patient’s immediate family (their parents and siblings). All of the above relies on strong support and coordination with third sector entities (NGOs).

By carrying out a qualitative diagnosis of needs based on the parents’ own perceptions, it has been possible to identify the priority issues that affect them. At the clinical level, these results are relevant to emphasize the role of interdisciplinary teams (including doctors, psychologists and social workers, among others). At the same time, this helps us to design intervention programs that respond to their difficulties, helping them adapt to the demands of the disease and achieve adequate levels of health and well-being.

As a limitation, it should be noted that it was very difficult to recruit the parents who finally participated, due to the sensitivity of the subject. This may be associated with the fact that, for most parents, undergoing an interview was a great emotional effort, in which they had to exercise their conscience and delve into circumstances in their lives that were especially painful for the child, the parents themselves and the rest of the family. At the same time, the language barrier (in the immigrant population) and families who declined the invitation to participate were other difficulties to be faced. In addition, the situation of the child, in terms of the treatment and care he needed (including the periods of “hospital leave”, between cycles of chemotherapy, which he spent at home), as well as the difficulty of the parents to reconcile this circumstance with the rest of their daily responsibilities, was reflected when trying to make an appointment for the session. In this sense, it had already been discovered that “childhood cancer” research was a difficult field in which to accumulate large samples [60].

## 5. Conclusions

This study has contributed to the existing literature by using a qualitative approach in analyzing the needs of families affected by childhood cancer, based on the parents’ own perceptions. It has supplemented the quantitative analysis already carried out on a larger sample [3,49].

The information gained from the interviews with parents has enabled us to respond to the question posed at the beginning: “what is the experience like of having a child with cancer?”. The answer is simple; parents are affected in all areas of their lives, in terms of the family, daily routines, work and household economics and, most importantly, their physical and psychological health. Fears and worries about the child occupy much of their existence, to the extent that they can experience serious health problems themselves.

In response to the question “In what areas do problems manifest most?”, the needs expressed by the parents themselves are clearly and fundamentally related to perceived social support, especially from informal sources. Similarly, the statements reflect their difficulties in coping with the situation created by their child’s illness; the coping strategies employed are not always the best for dealing with their fears and concerns (another of the areas indicated). The stressors they face are many and varied and it is evident how difficult it is to deal with them. The sample of parents who participated in this research point to the specific symptoms caused by the illness.

## Figures and Tables

**Table 1 children-09-01957-t001:** Overview of the interview questions (in order of appearance).

Topics	Central Questions
Introduction	Interviewer presentation and study objectives
Health (symptoms)	▪ Do you have any health problems, or have you suffered from any type of symptoms since your child’s illness or diagnosis? What are they? ▪ If applicable: what specialized healthcare or pharmaceuticals, etc., do you receive? ▪ Is any member of your family receiving psychological or psychiatric help? What type? ▪ If applicable: from your perspective, how relevant are your health issues in your daily life?
Stressors/Imbalances	▪ What problems and/or stressors do you experience from caring for your ill child? ▪ What problems and/or stressors do you experience from caring for the rest of your family? ▪ Which of the stressors mentioned above are the most common? ▪ If applicable: from your perspective, how relevant are these stressors in your daily life?
Coping	▪ In summary, how do you manage the stressful situations related to your child’s illness? ▪ If applicable: from your perspective, how important is it to know how to deal with these stressors?
Work/household economy	▪ What difficulties do you have in reconciling work and family life with your child’s illness? ▪ What changes have you experienced in your work life since your child’s diagnosis? ▪ If applicable: what type of financial problems are you experiencing? ▪ If applicable: from your perspective, how much impact do workplace obstacles and financial issues have on your daily life?
Education	▪ Has your child’s school performance been negatively affected by his/her illness (periods of hospitalization, lack of school attendance, learning problems, etc.)? ▪ As a result of the illness (side effects), has your child experienced learning problems? ▪ Has your child had (or is he/she having) problems with schooling?
Informal support	▪ Who has helped you most during your child’s illness? ▪ What changes have you experienced in the relationship with your partner since your child’s diagnosis?
	▪ Do you have other people close to you (friends, neighbors, co-workers, etc.) who are willing to help you on a day-to-day basis? ▪ Do you have relatives close by who can help you when you need it?
Formal support	▪ How would you define the support received from the doctors? ▪ And the support received from other healthcare workers (nurses and nursing assistants)?
Formal resources	▪ What is your opinion on the services available (both public and private) to support the needs of your child and your family in this situation? ▪ Is your child receiving any type of benefit/social care/program or help as a result of his/her illness? If yes, are you satisfied with it? Is it enough to cover your needs?
Fears and concerns	▪ What would you say are your main fears regarding the disease situation? ▪ And what are your main concerns regarding your child?
Others	▪ Of all the topics covered, which ones do you consider to be a priority? Why? ▪ Is there anything you want to tell me that is important to you but we have not yet talked about?

**Table 2 children-09-01957-t002:** Areas of need: Content Analysis Themes, Subthemes and Meanings.

Theme	Subtheme	Meaning *
Informal social support	Source of support: spouse, friends, neighbors and co-workers Isolated and dysfunctional support	Source and perception of support understood as being from the spouse, as well as from friends, neighbors and co-workers, and being satisfied with it.
Coping	Helplessness Active coping Negative coping	Classification of the different modes of coping expressed.
Fears and worries	Sequelae and relapse Family stability: future and happiness, the family as a whole Economic and work problems	Identification, prioritization and justification of the participants’ fears and concerns in relation to the pathology.
Stressors/imbalances	Disease claims (emotional effects) Exposure to medical procedures Combined disease/family	Circumstances/situations related to the disease that, in general, suppose an emotional imbalance or relate to family functioning.
Health problems in parents	Physical symptoms Psychological pattern Unhealthy routines and practices	Physical and/or psychological symptomatology and/or related aspects.
Other themes	Work and family economy, education of children, formal support and social resources	Other manifest demands or areas of their lives that have been negatively affected.

* Following Mayring [43], the theory of the definition (“meanings”) coming from the classic view of the categories, enumerating the sufficient conditions to belong to that category. Based on this explicit definition (studied and agreed upon by the authors), the sorting could take place.

## Data Availability

Not applicable.

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
