# Peer review of "Needs Assessment in Parents of Children Affected by Cancer: A Qualitative Perspective"

_children, 2022, doi:10.3390/children9121957_

Round 1
Reviewer 1 Report
The main question of the research is to analyze the needs of a group of parents who have a child diagnosed with cancer. The topic is original and relevant in this field. There is not much qualitative research in this area, and with the qualitative method, we can analyze more deeply the needs of parents of sick children. It would be useful to write more about the coding process and the codes. The conclusions are consistent with the evidence and arguments presented and address the main questions asked. The references are appropriate. The current tables are adequate. A summary table is required for the results.
The manuscript is important and well presented.
My recommendations:
1. Please shorten the introduction part
2. The examples are very expressive, but there are too many examples. Please add a summary table to the results to make it easier to see the topics.
3. Children and parents struggle with many psychosocial problems. Is there psychologist and/or social worker in the health care team to deal with the family's psychosocial problems? Please write more about it.
Author Response
Point 1.
Response 1: We have reduced the length of the "introduction" section.
Point 2.
Response 2: Examples of parenting sentences have been reduced in the text, incorporating the most significant ones into a summary table. Given the space limitations it is not possible to include all the answers.
Point 3
Response 3: we have taken the reviewer's suggestion into account. In fact, we already had a related reference.
Reviewer 2 Report
Solid and substantively important content that is also nicely linked to national policy and policy implications.
Author Response
Thanks for your considerationReviewer 3 Report
Thank you for the opportunity to review this well-written and rigorous qualitative study of parental needs during a child's cancer treatment. Overall, I found the paper easy to understand and rigorous. However, several revisions could make the paper much stronger.
Abstract: The abstract feels very vague with more presentation of methods than results. Also, I don't know what this sentence means: "Conciliation problems were also very real."
Introduction:
Page 2 - I dont understand this sentence: " Systematic needs assessment is a primary task." A primary task for whom? Also, in the research or clinical context? Please clarify.
Page 3 - The authors state that little is known about the experiences of parents of children with cancer, but they cite multiple studies in the introduction that explore just this topic. How is this study unique and needed?
Methods:
Page 4 - The authors refer to 'ad hoc questions'. Do they mean the "if applicable" questions listed in the interview guide? Or do they mean interviewers used prompts to follow novel thoughts during interviews?
Presumably, interviews were initially in Spanish. Please describe how they were translated.
Results:
Page 7 - The authors refer to the 'incidence' of themes. This is a qualitative study without a representative sampling technique. Therefore, I would avoid this type of quantification.
I found it unusual that authors included references in the presentation of the results. I believe results should be only the results, and these references and explications should be part of the discussion.
Page 7 - Theme 1. What does this sentence mean? "Concerning the spouse, the experiences cause feelings of dissatisfaction."
'Fears and Concerns' has a lot of overlap with 'Coping'. For example, excerpts from both themes describe parents feeling 'powerless' and the role of worry. Are these truly 2 different themes? If so, please spend time in the discussion explaining how these themes interact and overlap.
Discussion:
Pge 12 - The authors state they used 'previously validated scales' in these interviews. I don't know what this means. Scales generally refer to survey studies, not qualitative interviews.
Much of the discussion is restating the results, rather than exploring the meaning of the results. Perhaps add discussion on (1) What role clinicians can play in identifying and/or supporting these needs, (2) what we should do next, now that we know these needs exist, (3) Limitations of the current study.
Author Response
Introduction
Point 1.
Response 1: The changes indicated in the abstract have been made.
Point 2.
Response 2: the corresponding clarification has been made.
Point 3.
Response 3: The above literature shows that childhood cancer and its treatment lead to significant changes in the lives of caregivers of children with paediatric cancer, especially for parents. However, not much is known about the needs of parents from their own experience (qualitative approach). Most of the studies that analyse the psychosocial impact on parents, or the family as a whole, do so from a quantitative perspective, even developing specific inventories for evaluation.
Others, either from a quantitative or qualitative perspective, analyse a specific type of need, for example care needs or information needs. However, few works are aimed at knowing the areas of need in an integral way and from the parents' own perspective. It is they who define their problems and priorities. Thus, the results may help to develop or to optimize support services and interventions aimed at responding to the difficulties detected that help parents cope with the difficult situation of their children’s illness.
Finally, there are previous studies focused on the caregivers in general. These don’t have to be parents as caregivers.
For all these reasons, we consider this study pertinent, unique and needed, taking also into account that there are no previous studies similar to this population in Spain.
Methods
Point 4
Response 4: It is an interview guide designed by the researchers to measure, according to the information needs in this investigation.
Clarification on the language of the interviews:
The design of the interview and its administration to the participants was carried out in Spanish. This article has been translated by the Language Center of the University of Almería for submission to this Journal.
Results
Point 5
Response 5: the corresponding clarification has been made.
Point 6
Response 6: The incorporation of bibliography in the results section is done in an introductory way for each topic to be dealt with and in a descriptive way. It is not used for purposes of comparison and interpretation of results, as is the case in the discussion.
In any case, to avoid confusion, some of these references have been removed, keeping only those necessary for a better understanding of the results.
Point 7
Response 7: the corresponding clarification has been made.
Point 8
Response 8: These are actually two different themes; Table 2 describes the meaning of these areas of need independently because that is how it is determined by the MAXQDA application.
It is logical that there could be an overlap in the answers of the parents, especially when we ask them a question as open as their fears and concerns. Many of those fears will be related to other issues: coping, health, stress, even lack of support, etc.
Discussion
Point 9
Response 9: the corresponding clarification has been made.
Point 10.
Response 10: New paragraphs have been added in the Discussion section to respond to the reviewer’s suggestion
Round 2
Reviewer 1 Report
1. What is the main question that the research deals with?
The main question of the research is to analyse the needs of a group of parents who have a child diagnosed with cancer.
2. Do you think the topic is original or relevant in the field? Do it
address a specific gap in the field?
The topic is original and relevant in this field.
3. What does it add to the topic compared to other published materials?
With the qualitative method we can analyze more deeply the needs of parents of sick children.
4. What specific developments should the authors consider in terms of methodology? What additional checks should be considered?
The metodology is appropriate
5. Are the conclusions consistent with the evidence and arguments presented and do they answer the main question posed?
The conclusions are consistent with the evidence and arguments presented and address the main questions asked.
6. Are the references correct?
References are correct
7. Please attach additional comments to the tables and figures.
The tables are correct
Reviewer 3 Report
Thank you for responding to my comments.